# Exploring the Potential of Cold Sintering for Proton-Conducting Ceramics: A Review

**DOI:** 10.3390/ma17205116

**Published:** 2024-10-19

**Authors:** Andrea Bartoletti, Elisa Mercadelli, Angela Gondolini, Alessandra Sanson

**Affiliations:** Institute of Science, Technology and Sustainability for Ceramics (ISSMC) of the National Research Council (CNR), Via Granarolo 64, I-48018 Faenza, RA, Italy; angela.gondolini@issmc.cnr.it (A.G.); alessandra.sanson@issmc.cnr.it (A.S.)

**Keywords:** cold sintering, CSP, water-assisted densification, BZY, BCZY, yttrium-doped barium cerate-zirconate, ceramic proton conductors, hydrogen, low sintering temperature

## Abstract

Proton-conducting ceramic materials have emerged as effective candidates for improving the performance of solid oxide cells (SOCs) and electrolyzers (SOEs) at intermediate temperatures. BaCeO_3_ and BaZrO_3_ perovskites doped with rare-earth elements such as Y_2_O_3_ (BCZY) are well known for their high proton conductivity, low operating temperature, and chemical stability, which lead to SOCs’ improved performance. However, the high sintering temperature and extended processing time needed to obtain dense BCZY-type electrolytes (typically > 1350 °C) to be used as SOC electrolytes can cause severe barium evaporation, altering the stoichiometry of the system and consequently reducing the performance of the final device. The cold sintering process (CSP) is a novel sintering technique that allows a drastic reduction in the sintering temperature needed to obtain dense ceramics. Using the CSP, materials can be sintered in a short time using an appropriate amount of a liquid phase at temperatures < 300 °C under a few hundred MPa of uniaxial pressure. For these reasons, cold sintering is considered one of the most promising ways to obtain ceramic proton conductors in mild conditions. This review aims to collect novel insights into the application of the CSP with a focus on BCZY-type materials, highlighting the opportunities and challenges and giving a vision of future trends and perspectives.

## 1. Introduction

The EU’s hydrogen strategy and REPowerEU plan have put forward a comprehensive framework to support the uptake of renewable and low-carbon hydrogen to help decarbonize the EU in a cost-effective way and reduce its dependence on imported fossil fuels [1]. In this scenario, fuel cells and electrolyzers have the potential to meet the world’s current energy needs due to their efficiency, stability, and wide range of applications, from stationary to transportation sectors [2]. Electrochemical devices based on proton-conductive ceramic materials have gained increasing attention as an appealing solution due to the possibility of working efficiently at lower (intermediate) temperatures [3]. Recent studies have fully demonstrated the possibility of applying ceramic proton conductors in protonic ceramic fuel cells (PCFCs) and electrolyzers (PCEs) [4,5,6,7,8,9,10], membrane rectors [11,12,13,14], hydrogen purification devices [15,16,17,18], and hydrogen sensors [19,20].

Perovskite oxides are among the most promising ceramic proton conductors. Ideal perovskites have the general formula ABO_3_, where the A-sites are typically occupied by larger cations than the B-sites and analogous in size to the O-site anions. The ABO_3_ structure can be considered a face-centered cubic (fcc) lattice with the A atoms at the corners and the O atoms on the faces. The B atom is located at the center of the lattice. The structure of perovskite is formed by a three-dimensionally connected system of BO_6_ octahedra and AO_12_ cube-octahedra at the edges of the B-centered cubic lattice. In proton-conductive perovskites, a trivalent cation is generally partially substituted into the B-site to increase the oxygen vacancy concentration in the composite. The general formula of these doped perovskites can be written as AB_1−x_M_x_O_3−δ_, where x is less than the upper limit of its solid solution formation range (usually less than 0.2), and δ denotes the number of oxygen deficiencies per unit formula of the perovskite.

Perovskite materials, such as doped BaCeO_3_ (BCO), BaZrO_3_ (BZO), and their solid-solution BaCe_x_Zr_y_Y_1−xy_O_3−δ_ (BCZY)- and BaCe_x_Zr_y_Y_z_Yb_1−x−y−z_O_3−δ_ (BCZYYb)-type ceramics, have been extensively studied thanks to the good compromise among chemical stability, thermomechanical resistance, and high protonic conductivity. They are expected to play a pivotal role in the near-term development of performant, low-cost devices for the H_2_ economy [3].

The proton conductivity of these systems relies on the defect formation and distribution in the perovskite lattice, but also on external factors, such as temperature, partial pressure, and the nature of the atmosphere. Protons are formed in the water vapor or hydrogen-containing environment at high temperatures according to the following reactions (1)–(3):(1)$$H_{2}O+V_{O}^{∙∙}↔O_{O}^{x}+2H^{∙}$$
(2)$$\frac{1}{2}H_{2}+h^{∙}↔H^{∙}$$
(3)$$\frac{1}{2}H_{2}↔H^{∙}+e^{\prime }$$
where H^•^ stands for a proton. Moreover, it was found that protons bond with oxygen atoms to form substitutional hydroxyls, as follows [21]:(4)$$H_{2}O+V_{O}^{∙∙}+O_{O}^{x}↔2OH_{O}^{∙}$$
(5)$$H_{2}+{2O}_{O}^{x}+2h^{∙}↔2OH_{O}^{∙}$$
which are the main reactions responsible for proton conduction. Thus, for protonic conduction, the material first needs to incorporate protons through hydroxide formation, and then the transport mechanism takes place by interstitial proton hopping through intra-octahedral O sites in the presence of oxygen vacancies [22]. Proton migration through the lattice via the hopping process includes the fast rotation and reorientation of the proton, which implies the localization of the proton in the vicinity of an oxygen ion, followed by its transfer to the next ion [23,24].

Since their discovery in the 1980s, several efforts have been made in the development of proton-conducting ceramic materials [25]. However, the manufacturing of these devices has faced several issues due to the poor sinterability of cerate and zirconate perovskites, which requires high temperatures of up to 1700 °C and long processing times (4–40 h) to be fully densified (Table 1). The obtainment of hermetic gas-tight ceramic electrolytes is a key step to enhancing the proton conductivity and durability of PC-SOCs/SOEs [26]. Moreover, the refractive nature of this class of materials (especially BaZrO_3_-type ceramics) leads to the generation of a higher density of grain boundaries, which are generally more resistive compared to the bulk due to the existence of a space charge depletion layer that generates the so-called “grain boundary blocking effect” [27]. Recent studies have revealed that the proper optimization of the dopant concentration and processing methodology (manufacturing and sintering treatment) can effectively suppress the blocking effect, thus providing high proton conductivity [28].

An important aspect to consider when sintering BaCeO_3_-BaZrO_3_-type perovskites is their poor thermal stability at high temperatures. The first studies on this phenomenon were conducted by D. Shima and co-workers, who investigated the influence of cation non-stoichiometry on the conductivity of doped and undoped BaCeO_3_ ceramics [29]. They found that small changes in Ba content have a dramatic effect on proton conductivity and hypothesized that severe barium loss occurs during high-temperature sintering. This was further confirmed by Glockner et al. [30], who performed atomistic simulations of defect formation in BaCeO_3_ compounds, revealing the favorable formation energy for Ba and O vacancy pairs, which could result in the loss of BaO at very high temperatures. Experimental studies revealed similar behavior for the more stable, doped and undoped BaZrO_3_ ceramics [31,32] and BaCeO_3_-BaZrO_3_ solid solutions [33,34,35,36].

**Table 1 materials-17-05116-t001:** Sintering conditions and relative density of BaCe_x_Y_1−x_O_3–δ_ (BCY), BaZr_x_Y_1−x_O_3–δ_ (BZY), BaCe_x_Zr_y_Y_1−x−y_O_3–δ_ (BCZY), and BaCe_x_Zr_y_Y_z_Yb_1−x−y−z_O_3−δ_ (BCZYYb) ceramics considering different powder synthesis methods, such as solid-state synthesis and wet chemical approaches; values in bold indicate that sintering additives are used.

Material	Synthesis Route	Sintering Temperature (°C)	Sintering Time (h)	Relative Density (%)	Ref.
BaCe_x_Y_1−x_O_3−δ_	Solid-state synthesis	1450	40	90	[37]
		1550	10	92	[38]
		1675	10	90	[39]
		1500	6	70	[40]
	Wet chemical approaches	1500	6	85	[40]
		**1500**	**6**	**95**	[40]
		1350	10	74	[41]
		1250	10	63	[41]
		1650	4	90	[42]
		1250	10	91	[43]
		1400	10	96	[44]
BaZr_x_Y_1−x_O_3−δ_	Solid-state synthesis	1535	12	94	[45]
		**1325**	**20**	**96**	[46]
		1600	24	94	[47]
		1500	24	30	[48]
		1485	12	94	[45]
	Wet chemical approaches	1300	4	60	[49]
		1700	12	90	[50]
		1600	5	79	[51]
		1600	12	91	[52]
		**1500**	**10**	**90**	[53]
		1700	4	85	[49]
BaCe_x_Zr_y_Y_1−x−y_O_3−δ_	Solid-state synthesis	**1400**	**8**	**97**	[54]
		**1600**	**10**	**97**	[55]
		1600	24	96	[55]
	Wet chemical approaches	1450	5	80	[56]
		**1450**	**5**	**96**	[56]
		1550	4	90	[57]
		1450	10	89	[36]
		1450	5	88	[58]
BaCe_x_Zr_y_Y_z_Yb_1−x−y−z_O_3−δ_	Solid-state synthesis	1550	10	97	[59]
		1550	10	97	[60]
		**1450**	**3**	**98**	[61]
	Wet chemical approaches	1400	6	95	[62]
		1400	10	98	[63]
		**1400**	**5**	**97**	[64]

To overcome this issue, several efforts have been made to produce these ceramic proton conductors in mild conditions. As shown in Table 1, the use of powders produced by wet chemical approaches, such as co-precipitation and combustion synthesis, generally helps in lowering the sintering temperature and/or time with respect to the use of powders synthesized by the conventional solid-state methodology [65]. This is mainly due to the finer and more reactive nature of the powder produced by wet chemical strategies. Furthermore, the use of sintering aids such as ZnO, NiO, and CuO has been extensively studied and has proven to be effective in reducing both the sintering temperature and processing time, regardless of the synthetic approach [66]. In this case, the formation of a low-temperature melting phase is responsible for increasing the ion diffusion and the formation of defects, which both help mass transport and densification [67]. However, the sintering conditions are still quite harsh (with temperatures higher than 1300 °C) and highly energy-consuming. Moreover, the possible detrimental effect on the conductivity of proton-conductive ceramics due to the presence of residual sintering aids on the grain boundaries is still under debate [68].

Regardless of the sintering treatment, one well-consolidated strategy to repress barium evaporation from BaCeO_3_-BaZrO_3_ ceramics is to control the sintering atmosphere by introducing sacrificial Ba-containing powders (the so-called “pack”) or pellets in the sintering setup. In this way, it is possible to finely control the composition of the perovskite phase, even at a high temperature of about 1550 °C [26,34,69]. Otherwise, since BaO evaporation starts from the surface and only a small fraction of the superficial part of the material undergoes compositional changes, light mechanical abrasion of the surface of the sintered ceramics is generally carried out. However, this approach is suitable only at the lab scale for relatively thick devices; therefore, conventional PC-SOC/SOE devices produced with thin-film electrolytes and at medium–large scales have been excluded.

Although several improvements have been made in manufacturing proton-conductive ceramic materials, producing highly efficient devices in mild conditions is still a great challenge. Moreover, recent studies revealed the formation of Stacking Fault defects in BZY- or BCZY-type ceramics when conventionally sintered, and these defects detrimentally affect both the conductivity and mechanical properties of the final device [70]. Exploring novel sintering technologies could be an effective way to achieve technological advances in the production of highly efficient proton conductor devices.

Among others, the cold sintering process (CSP) presents unique advantages in terms of material properties, flexibility, and costs, and it is considered one of the most promising ways to obtain ceramic proton conductors in mild conditions. However, to the best of our knowledge, no comprehensive article has been published that provides a complete overview of the CSP findings and advancements in BaCeO_3_/BaZrO_3_-based proton conductors. For these reasons, the aim of this review is to provide an overview of the theory and technological advancements related to the cold sintering process. A comparison of various studies on BaCeO_3_/BaZrO_3_-based perovskites is also considered, highlighting the main challenges and advantages in the process application. The future trends and prospects of the method in this field are addressed as well.

## 2. Cold Sintering Process

Cold sintering is a geologically inspired sintering process where densification is driven by mechanical–chemical effects (pressure solution creep or dissolution–precipitation creep) in synergy with chemical effects [71].

Pressure solution is a fluid-assisted, stress-driven mass transport enabled by chemical potential gradients, typically associated with the upper Earth’s crust rock’s densification and deformation [72]. Despite being a very slow process in nature, it is strongly accelerated in fine-grained materials and can occur not only in the presence of aqueous solutions but also with a partial eutectic melt or any intergranular solution phase, as long as the grain boundaries are wetted [73].

During cold sintering, materials are densified in the presence of a transient liquid phase, a compound that promotes dissolution and precipitation reactions and subsequent compaction with the aid of an external uniaxial applied pressure (100–1000 MPa), and consolidation is carried out between room temperature and above the boiling point of the liquid (typically < 350 °C). The time required to obtain high-density ceramic materials is typically less than one hour [74].

A schematic representation of the cold sintering process and the thermocompression apparatus employed is shown in Figure 1. First, an appropriate amount of the transient phase (liquid or solid) is introduced to the particle ensembles. When the transient phase is a liquid, this first stage enables the local dissolution of the sharp surfaces of the particles, and the liquid acts as a lubricant to promote particle rearrangement and sliding. With the assistance of an applied external pressure, the phase fills the interstitial space among particles, providing the initial particle compaction. The temperature is raised to the boiling point (or melting point if solid) of the transient phase, enhancing the solubility of ceramics and the formation of a supersaturated environment through the evaporation of the liquid phase from the pellet and the die, which is not perfectly sealed.

The presence of an external uniaxial load provides the strain that promotes mass transport and densification. The driving force during the process is the chemical potential gradients, from highly constrained areas with enhanced dissolution and high chemical potential to weakly constrained areas at particle surfaces with a lower chemical potential, through the liquid film (Figure 2) [75].

A wide range of materials have been successfully densified through the CSP, including structural ceramics [73,76,77,78], microwave dielectrics [79,80,81,82,83], ferroelectrics [84,85,86,87,88], piezoelectrics [89,90,91,92], Li-ion cathodes [93,94,95,96,97], transparent ceramics [98,99,100,101,102,103], solid-state electrolytes [104,105,106,107,108,109,110,111,112,113,114,115], thermoelectric materials [116,117,118,119], ceramics for electronics [120,121,122] and environmental applications [123,124,125], supercapacitors [126], magnetic ceramics [93,127,128,129,130,131], bioceramics [132,133,134,135,136], bulk van der Waals materials (MXene, MoS_2_, reduced graphene oxide, etc.) [137], and metals [138,139]. Moreover, the mild sintering conditions made possible the co-sintering of dissimilar materials with large differences in processing temperature windows, such as new ceramic–polymer composites [140,141,142,143,144,145,146], and thermodynamically unstable phases [147]. The CSP is suitable for densifying materials not only in the form of powders but also in green ceramic parts (generally after organic removal) and multilayers. The conventional sintering and assembly of a working device require several steps with repeated heating, including forming (e.g., tape casting), multilayer assembly, firing, metallization of electrodes, and integration with a substrate. The CSP allows, in a single step, the integration of organic and inorganic components, and it has the potential to surpass conventional processing routes by offering unprecedented “all in one step” solutions to manufacture devices.

Recently, the CSP has been applied to the densification of barium cerate and zirconate ceramics. Cold sintering was successfully demonstrated for the fabrication of BZY- [112,148] and BCZY-based proton-conductive electrolytes [110,111,113,149,150], as well as cer–met anodes [151], for PC-SOC/SOE devices at the lab scale.

However, although several years have passed since the first patent in 2016, the studies related to cold sintering have mainly been devoted to the densification of powders. Despite the ease of operation and the simplicity of the equipment required for the cold sintering process (i.e., a thermocompression die apparatus), there are several parameters to be taken into account when considering the CSP (Figure 3), and the obtainment of compact ceramics with high density, phase purity, and engineered microstructure is not trivial. Further insight into the mechanisms involved during cold sintering is provided in the following paragraph.

## 3. Cold Sintering Mechanisms

The cold sintering kinetics and mass transport mechanisms are in the early stages of investigation, and a complete analytical model for a deep understanding of the sintering process is still lacking. However, experimental and theoretical investigations allow the identification of two main stages during the CSP (Figure 4). The following explanation does not take into account solid transient phases such as hydrated salts or eutectic mixtures.

Stage I involves interactions between ceramic particles and a liquid under the application of uniaxial pressure. When a solid substance is in contact with a liquid, four different phenomena can be observed: (i) dissolution, (ii) chemical reactions, (iii) absorption limited to the surface, (iv) electrostatic/steric repulsion/wetting. The first two are considered beneficial in promoting consolidation in the CSP [153]. Frequently, the liquid fraction is small (<10%), and consolidation is achieved by partial solubilization [154]. During Stage I, the total volume of the system slightly decreases because of drying, causing material shrinkage. This phenomenon takes place until the solid particles form a rigid skeleton, and then Stage II starts. This stage is predominated by pressure- and temperature-assisted dissolution and precipitation events that are driven by local and global gradients within the pellet die [155], promoting rapid densification through the pressure solution creep (PSC) mechanism. PSC is a dissolution–precipitation process that relies on the transport of matter from the contact between touching particles to the surrounding liquid phase and eventually to nearby non-contacting surfaces. Such transport locally decreases the distance between the particle centers, enabling global shrinkage and densification [73]. This mechanism is driven by chemical potential gradients, from highly constrained areas with enhanced dissolution and high chemical potential to weakly constrained areas at particle surfaces with a lower chemical potential, through atomically thin liquid films at the contacting particle surfaces [72]. At this stage, the system is likely to respond in at least three possible routes [155]: (1) the heterogeneous nucleation of dissolved species, a process that minimizes free energy by reducing the surface area and by ultimately replacing the solid–liquid interface with solid–solid grain boundary interfaces; (2) homogeneous nucleation, where new crystals precipitate in the interstitial space between grains [156]; (3) a step-wise transition, whereby a metastable glass phase or intermediate compound is formed to bridge the initial solutes and final product [157]. Recent studies have evidenced that, during this stage, most of the fluid is extracted from the pellet die [158]. At the end, materials are, in general, almost fully densified and are then subjected to constant temperature and pressure for several minutes, where the prevalent effect is grain growth, occurring with a relatively slow kinetic process [159].

During these stages, the transient chemistry, the sintering temperature and rate, uniaxial pressure, and dwell time (t) are the main densification process variables. Depending on the chemistry of the particle–solvent interaction, either congruent or incongruent dissolution on the surface can occur [154,160]. In the former case, the material dissolves into the solvent with a homogeneous chemical stoichiometry before mass transport and precipitation [127], and the resulting liquid solution enables the hydrothermal environment for precipitation and crystal growth [71]. The particles then precipitate from the supersaturated solution when a strain, temperature, or curvature gradient is encountered, or due to the evaporation of the solvent phase through the die [160]. Typically, when evaporation is fast, solute gelation and the subsequent formation of a disordered phase within the grain boundaries are observed [79,122,161].

Incongruent dissolution results instead in a material that has a different composition/stoichiometry when compared with the parent phase [71]. Here, ions with higher dissolution kinetics are leached out before those with slower dissolution rates, leading to the formation of unwanted secondary phases and hindering densification. Moreover, these byproducts are generally deleterious to the final properties [157,162,163]. The formation of undesired phases can be mitigated by saturating the transient phase with cations with higher solubility in the selected solvent [110,112,113,164] to slow down the dissolution kinetic of these ions, reducing the surface passivation process and allowing the dissolution–precipitation reactions to proceed. However, when incongruent dissolution occurs, it is almost impossible to completely prevent the formation of crystalline or amorphous/glassy phases within the sample. In such cases, a post-annealing (PA) step is required to induce solid-state reactions between crystalline impurities or speed up the metastable glass-to-crystalline phase transition. In these conditions, epitaxial crystal growth takes place via mass transport since the corresponding ionic species, atomic clusters, or ligands in the glass phase precipitate onto the surface of the crystallites (recrystallization) [71,165]. Post-annealing is often employed even when no secondary phases are present to increase the grain size, which is generally limited during cold sintering. It should be mentioned that, in some cases, the formation of secondary phases during cold sintering is desired and can be an effective way to engineer the grain boundaries of the material [149].

It has recently been shown that incongruent dissolution can be avoided by the use of more chemically active solvents, such as molten hydroxides and salts [162,166,167] or chelating agents like acetylacetonates [93]. This means that the selection of an effective transient chemical phase is not trivial and is probably the most influential parameter affecting the process.

It is worth mentioning that, in some cases, the transient phase was found to not dissolve ceramic particles (the case of negligible dissolution), but densification can also occur during cold sintering if soluble salts are mixed with the ceramics [168,169]. In such cases, the soluble phases lubricate the insoluble ceramics, resulting in compaction via plastic deformation or subcritical crack growth processes [170], but generally, a post-annealing step is required to further improve densification.

## 4. The Choice of the Transient Phase

As mentioned above, there are several factors affecting densification during the cold sintering process (see Figure 3), but the most critical parameter is certainly the choice of the right transient phase, which enables dissolution–precipitation reactions.

Since, during the CSP, hydrothermal reactions are accentuated, to adequately tackle the cold sintering process, it is of pivotal importance to be guided by predominance–existence diagrams (PEDs) [71]. These diagrams collect information on the stability of all the involved compounds, intermediate chemicals, and ionic species as a function of the solution concentration and pH environment, as well as the nature of the thermal treatment atmosphere. In this way, it is possible to control the dissolution chemistry and provide high densification without incurring undesired incongruent dissolution phenomena. Pourbaix diagrams are also useful tools to understand and predict mechanisms involved during CSP.

The transient phases employed in cold sintering can be divided into aqueous and non-aqueous. Regarding the first class, pure water, aqueous solutions containing dissolved or suspended metal precursors, organic acids, chelating agents, and hydrated salts have been successfully employed to drive the cold sintering process. To avoid the formation of undesirable hydroxides, the use of non-aqueous liquids in the CSP is an option. Several polar and semi-polar solvents have been employed in cold sintering since the dissolution of inorganic compounds is promoted by the higher relative permittivity of the liquid phase. Moreover, eutectic salt mixtures have been widely employed as a “high-temperature solution” to promote densification during the CSP [171]. In some cases, aqueous/non-aqueous mixtures were also investigated. The choice of the transient phase is typically related to the solubility of the ceramic particles in the solvent but also depends on the cold sintering equipment (i.e., the material of the die). Generally, establishing the proper transient phase, including the nature and concentration of the solutes and pH value, enables a favorable environment for chemical reactions and reduces the activation energy for mass transport in the initial stages of the CSP [172].

A list of some possible aqueous and non-aqueous transient phases that can be used to densify different classes of materials, as well as the relative cold sintering parameters, is reported in Table 2.

### 4.1. The Modification of the Crystal Structure and Chemical Composition During Cold Sintering

In all the cases described in Table 2, the initial powders and the final dense part have the same chemical compositions and crystallographic phases. However, it is possible to apply cold sintering to induce allotropic changes or chemical reactions. Changes in chemical compositions can be obtained by dehydration/decomposition reactions via a “precursor” approach, where the starting powder loses inorganic functional groups and/or chemisorbed water to form the most thermodynamically stable oxide under the applied conditions [127]. For example, Ndayishimiye et al. obtained α-Fe_2_O_3_ from α-FeOOH (“precursor”) and FeCl_3_ (“transient phase”) at 310 °C and 530 MPa. Kang et al. [173] produced high-density (≈97%) SiO_2_ glass at 300 °C and 300 MPa of applied pressure starting from H_2_SiO_3_ precursors with a 5 M NaOH solution. Bhoi et al. [174] obtained ≈90% dense SnO starting from a Sn_6_O_4_(OH)_4_ precursor at 270 °C and 300 MPa. Regarding allotropic changes during cold sintering, Floyd et al. [175] reported the phase transition of Li_2_MoO_4_ from phenacite to spinel under cold sintering in the presence of 6 wt. % H_2_O at 120 °C and 700 MPa for 30 min. Furthermore, Yamaguchi et al. [176] observed the transition from gibbsite to boehmite when cold sintering aluminum hydroxide powders at 200 °C and 270 MPa for 60 min using 6 wt. % of water.

Another way to change the chemical composition during cold sintering is to perform so-called “reactive cold sintering”, which takes place when a mixture of precursors that can react together under the applied conditions is employed to yield a new material or a doped compound [127]. Sada et al. [177] investigated, for the first time, reactive cold sintering by using Sr(OH)_2_·8H_2_O as the transient flux to assist the densification of BaTiO_3_ ceramics at 80 °C and 350 MPa [105]. In these conditions, Sr(OH)_2_·8H_2_O acts as both the transient phase and doping compound, leading to the formation of BaTiO_3_-Ba_1−x_Sr_x_TiO_3_ ceramics. A similar approach was employed by Ndayishimiye et al. [127], who obtained dense zinc ferrite (ZnFe_2_O_4_) ceramics from α-FeOOH and Zn(OH)_2_ precursors in the presence of NaK eutectic flux at 395°C and 530 MPa. Guo et al. [178] obtained dense (≈97%) BaTiO_3_ ceramics by treating Ba(OH)_2_·8H_2_O and H_2_TiO_4_ starting powders at 350 °C and 500 MPa for 120 min. The authors found that similar reaction pathways can also be used to densify SrTiO_3_ and Ba_1−x_Sr_x_TiO_3_ ceramics. Moreover, Zhao et al. [108] obtained Gd-doped CeO_2_ ceramics starting from Ce(NO)_3_·6H_2_O and Gd(NO)_3_·6H_2_O by a cold sintering + annealing process, achieving 94% dense Gd_x_Ce_1−x_O_2−δ_ parts after post-heat treatment at 950 °C. The low-temperature process results in no phase segregations along the grain boundaries, leading to exceptional grain boundary conductivity values (10 times higher compared to the conventional sintering temperature).

**Table 2 materials-17-05116-t002:** A list of possible transient phases to densify different types of ceramics though the cold sintering process.

Transient Phase	Densified Materials	Cold Sintering Parameters			Density (%)	Ref.
		T (°C)	P (MPa)	t (h)		
Aqueous solutions						
Water	V_2_O_3_	120	350	0.25	90	[74]
	K_2_Mo_2_O_7_	120	350	0.25	94	[74]
	KH_2_PO_4_	120	350	0.02	98	[71]
	NaCl	20	300	0.17	99	[170]
	Bulk hBN	45	375	0.17	99	[137]
NaOH solution	WO_3_	160	530	0.5	92	[179]
Ba(OH)_2_/TiO_2 (aq)_	BaTiO_3_	180	430	0.5	90	[71]
Zn(CH_3_COO)_2 (aq)_	ZnO	250	150	0.1	97	[180]
Acetic acid	ZnO	125	355	1	95	[181]
Formic acid	ZnO	125	355	1	95	[179]
Citric acid	ZnO	125	355	1	80	[179]
ethylenediaminetetraacetic acid	Fe_3_O_4_	350	430	1	80	[127]
	Li_2_TiO_3_/PTFE	150	350	0.5	95	[182]
Acetylacetonate	Ni-Cu-Zn ferrite	300	1000	0.75	93	[93]
	ZnO	140	350	2	96	[183]
Ba(OH)_2_∙8H_2_O	BaTiO_3_	225	350	1	95	[184]
Zn(CH_3_COO)_2_∙2H_2_O	ZnO	120	530	0.5	97	[183]
Non-aqueous solutions						
Dimethyl formamide (DMF)	Li_6.95_Mg_0.15_La_2.75_Sr_0.25_Zr_2_O_12_–[CH(CH_3_)CH_2_OCO_2_]_n_–LiClO_4_	120	400	1.5	90	[185]
Ethanol	HBO_2_	120	500	0.17	95	[186]
	Li_1.5_Al_0.5_Ge_1.5_(PO_4_)_3_	120	400	0.33	80	[163]
NaOH-KOH 51:49 eutectic mixture	BaTiO_3_	300	520	12	96	[166]
	K_x_Na_1−x_NbO_3_	300	520	0.5	94	[187]
Aqueous/Non-aqueous mixtures						
Dimethyl sulfoxide (DMSO)–acetic acid (9:1)	MnO	250	530	1	94	[156]
	ZnO	180	530	1	98	
DMF–water (1:1)	Li_1.5_Al_0.5_Ge_1.5_(PO_4_)_3_–LiC_2_F_6_NO_4_S_2_	150	370	1	87	[188]
DMSO–water–acetic acid (3:6:1)	ZnO/MoS_2_	170	480	0.75	>90	[189]
NaOH-KOH 51:49 moisturized eutectic mixture	ZnO	200	90	0.5	96	[89]
	Bi_2_O_3_	200	530	0.5	97	
	K_x_Na_1−x_NbO_3_	200	530	0.5	94	
LiNO_3_-LiOH 60:40 moisturized eutectic mixture	Li_1.3_Al_0.3_Ti_1.7_(PO_4_)_3_	200	400	1.5	>90	[190]

### 4.2. Other Parameters Affecting the Cold Sintering Process

In addition to the transient phase, the temperature and the pressure applied during the process, the heating rate, and the starting particle size are parameters affecting the solubility of ceramic particles in the solvent; therefore, they have to be carefully evaluated to create a suitable environment for densification. A study conducted on ZnO ceramics showed that the atmosphere applied during sintering (vacuum, argon atmosphere, air) seems to not influence densification or grain morphology during the CSP [191]. However, this should also be proven for other classes of materials before excluding an active role for this parameter in the process.

#### 4.2.1. Temperature and Pressure

Typically, the temperature is increased up to the boiling point of the transient phase to enhance the solubility of the ceramic particles and create a hydrothermal environment with the assistance of a few hundred megapascals of applied pressure. It is worth mentioning that reaching the hydrothermal environment is not a universal requirement to perform cold sintering [192].

High pressure is generally preferred since it enhances particle compaction and can trigger other densification mechanisms, such as plastic flow. Moreover, recent observations suggest that (i) the rates of densification and coarsening depend on the uniaxial pressures applied [193] and (ii) there is a critical pressure (~1 GPa for ZnO) associated with the transition from heterogeneous to homogeneous nucleation during recrystallization [194]. When the pressure is lower than this critical value, the applied pressure enhances densification and grain growth, while above the critical value, densification is dramatically retarded, and many small crystals are formed in the interstitial space between grains.

Hydrothermal (or solvothermal) conditions usually refer to high pressure and a relatively high temperature of aqueous or non-aqueous solutions to promote heterogeneous reactions such as dissolution and recrystallization. The equilibrium vapor pressure of water can be expressed with the Clausius–Clapeyron equation (Equation (6)):(6)$$\frac{dP}{dT}=\frac{∆H}{T∆V}$$
where Δ*H* and Δ*V* are the latent heat and the specific volume change of the phase transition. As the temperature increases, the pressure of liquid water increases significantly, reaching several hundred MPa at mild temperature ranges. These conditions lead to strong changes in the properties of the solvent. Regarding water, which is the most used transient phase, (i) the dielectric constant decreases drastically with increasing *T*, and the dissolution of non-polar compounds will become more favorable; (ii) the water’s ionization product (Kw) changes, making water more highly ionized and thus speeding up chemical reactions, i.e., dissolution and precipitation [195]; (iii) water becomes less viscous, and the mobility of ions is significantly enhanced [196].

#### 4.2.2. Heating Rate

Generally, there are two main heating regimes applied during cold sintering. Slow heating rates of ≈5–20 °C/min are beneficial for the elimination of surface-desorbed species (mainly carbonates), which behave as a barrier to volume diffusion from the crystal bulk toward its surface. When water is used as a solvent, the replacement of surface carbonates by hydroxides was found to contribute positively to densification [191].

Fast heating rates (>100 °C/min) are typically applied in field-assisted sintering technology/spark plasma sintering (FAST/SPS) facilities when the amount of transient phase is very low (<1.5 wt. %) or when hydrated salts are used. A recent study performed by Jabr et al. [197] demonstrated that the heating rate can influence the formation of defects during the process. Fast heating rates can induce pressure build-up inside the sample and hinder diffusion, leading to defects and inhomogeneities in the whole sample.

In general, an optimal heating rate should provide a balance between the kinetics of pressure solution creep and liquid phase evaporation throughout the specimen. It should be mentioned that the optimal heating rate is strongly influenced by the technical specifications of the thermocompression die used.

#### 4.2.3. Particle Size of Raw Materials

The use of nanoparticles instead of coarse particles is another way to enhance the solubility of the material since the relation between the solubility of the solid phase in the liquid and its particle size follows the Ostwald–Freundlich equation [66], given in Equation (7):(7)$$ln\frac{S_{r}}{S_{\infty }}=\frac{2\Omega_{A}\gamma_{sl}}{kTr}$$
where *S_r_* is the solubility of a particle with radius *r*, *S*_∞_ is the solubility of a particle with an infinite radius, *γ_sl_* is the interfacial tension, Ω*_A_* is the atomic volume, *k* is the Boltzmann constant, and *T* is the absolute temperature. In addition, nanoparticles provide more lattice sites for the dissolution process and nucleation during precipitation due to the high surface-to-volume ratio [157,172]. However, the use of fine mono- or polydisperse nanopowders (≤50 nm) was found to be detrimental to compaction during cold sintering [191,198]. The reason is not perfectly clear but might be related to the greater tendency of small nanopowders to aggregate during the mixing step with the transient phase (causing inhomogeneous flux distribution during CSP), coupled with the higher stress generated in the die when finer particles are pressed, which increases the possibility of capping and delamination during cold sintering.

From a thermodynamic perspective, better results are obtained in the CSP when using bimodal-sized nanoparticles [84,90,91,191]. Mixing nanoparticles into a “bimodal particle size composite” creates a thermodynamically unstable system because of the nonuniform surface energy/stress environment, and the local energy/stress gradient enables a driving force for the Ostwald ripening crystal growth process to force the entire system to reach an equilibrium state by minimizing the surface energy [157]. It is worth mentioning that the pressure solution creep mechanism can avoid the limits of constrained sintering encountered in traditional sintering approaches when bimodal particles are used [198]. Moreover, the use of starting powders with a bimodal size distribution generally enhances the flowability and compaction of particles during the first stage of the process (similar to conventional dry pressing technology).

## 5. The Application of the Cold Sintering Process to Ceramic Proton Conductors

The feasibility of performing the CSP on barium cerate and/or zirconate ceramics has been demonstrated over the years, and the results obtained up to now are reported in Table 3. The main strategy adopted to cold-sinter BCZY- and BZY-type ceramics involves the introduction of an aqueous transient phase into the as-synthetized ceramic powders.

Kindelmann et al. [148] cold-sintered BaZr_0.8_Y_0.2_O_3−δ_ (BZY20) ceramics starting from BaCO_3_, ZrO_2_, and Y_2_O_3_ precursors planetary ball milled in isopropanol and calcined at 1175 °C for 3 h in air. CSP experiments were performed in a FAST/SPS machine in the presence of 5 wt% deionized water as the transient phase. Experiments were carried out at temperatures between 150 and 250 °C, applying 400 MPa of uniaxial pressure and a dwelling time between 10 and 60 min. The relative density increased from 80 to 85 and 87% when increasing the temperature from 150 to 200 and 250 °C, respectively. At the considered maximum temperature of 250 °C and 400 MPa, cold-sintered BZY20 exhibits a heterogeneous microstructure with the presence of residual pores and secondary phases. When cold sintering is performed in BZY-type compounds, densification is driven by the formation of Y(OH)_3_ species [148] as follows:(8)$$Ba(Zr,Y)O_{3}+H_{2}O\to \left(Zr\;rich\right)Ba(Zr,Y)O_{3}+Ba(OH{)}_{2}+Y({OH)}_{3}$$

Since cold sintering is generally performed in an ambient atmosphere (i.e., CO_2_ is present in the reaction environment), barium hydroxide rapidly reacts with the dissolved CO_2_ to form the more thermodynamically stable barium carbonate:(9)$$Ba(OH{)}_{2}+CO_{2}\to BaCO_{3}+H_{2}O$$

It is worth mentioning that BaCO_3_ was also found when cold sintering was performed in a controlled atmosphere [148], but the presence of BaCO_3_ in the starting powders could be an explanation. Moreover, barium-based compounds react easily with the atmospheric CO_2_ according to
(10)$$Ba(Ce,Zr,Y)O_{3-\delta }+CO_{2}⇆Ba_{1-x}(Ce,Zr,Y)O_{3-\delta -x}+xBaCO_{3}$$

Interestingly, Kindelman et al. [148] pointed out that pressure solution creep mechanisms are not activated by the presence of BaCO_3_ alone, since no cold sintering activity was observed in BaZrO_3_ ceramics processed in the same conditions as BZY.

Considering the high amount of secondary phase produced, a post-annealing treatment (PA) was required to improve the densification degree and phase purity. Here, a *two-step* approach was considered, and a thermal post-treatment between 900 and 1100 °C in the same FAST/SPS apparatus imposing 100 MPa of uniaxial pressure was performed directly after cold sintering. This innovative approach allowed the formation of highly dense (≈95%) BaZr_0.8_Y_0.2_O_3−δ_ with nano-sized microstructure but with evidence of impurities such as Y_2_O_3_ and BaCO_3_ that detrimentally affect the proton conductivity of the produced material. The conditions employed for the PA treatment should be carefully evaluated based on the thermodynamic stability of the compounds formed during cold sintering.

Zhao et al. [112] cold-sintered BZY20 ceramics starting from as-synthesized powders moisturized with 20 wt. % of an aqueous PVA solution (3 wt. %) at 180 °C and 400 MPa for 1 h. A high green density of up to 76% was achieved after cold sintering, which is about 20% higher than the relative density obtained after traditional dry pressing. However, when a diluted polyvinyl alcohol (PVA) solution is used instead of pure water [112], some polymeric residue could remain in the green pellet since the boiling point of PVA is around 340 °C (melting point 200 °C). X-ray diffraction analysis revealed the presence of BaCO_3_ impurities, but no evidence of Y(OH)_3_ was detected, likely due to its low concentration or poor crystallinity, making it difficult to detect using this technique. Here, a post-annealing treatment at 1500 °C for 12 h was performed to reintroduce these secondary phases into the crystalline lattice. In this way, highly dense (94%) and pure-phase BZY20 ceramics were obtained, showing a homogeneous microstructure composed of micrometric grains and electrochemical properties comparable to those of traditionally sintered samples thermally treated at higher temperatures of up to 1700 °C.

Another feature of the cold sintering process is the possibility of producing the desired ceramic phase in situ (i.e., directly in the sintering environment) through reactive cold sintering. Shen et al. [199] obtained pure BaZrO_3_ ceramics with an acceptable density (≈92%) after treating Ba(OH)_2_·8H_2_O and Zr(OH)_4_ precursors at 300 °C and 500 MPa for 2 h. Here, densification is carried out by the dehydration of Ba(OH)_2_·8H_2_O, followed by the dissolution of barium hydroxide and the formation of an alkaline environment, as follows:(11)$$Ba(OH{)}_{2}∙8H_{2}O\overset{∆}{\to }Ba(O{H)}_{2}+{8H}_{2}O$$
(12)$$Ba(OH{)}_{2}\overset{H_{2}O}{\to }Ba^{2+}+2OH^{-}$$

As the pH increases, the solubility of zirconium hydroxide rises, and BaZrO_3_ nuclei are rapidly formed with increasing temperature, while the holding time promotes the growth and maturation of BaZrO_3_ particles. Although no information is provided regarding the performance of the reactive cold-sintered BaZrO_3_ ceramics, this method can be an interesting way to obtain dense and pure proton-conductive perovskites in one step, avoiding post-annealing treatments.

Regarding Ce-containing compositions (i.e., BCZY-type), Thabet et al. [110,111] cold-sintered as-synthesized BaCe_0.8_Zr_0.1_Y_0.1_O_3−δ_ powders obtained by the nitrate–glycine method at temperatures from 120 to 180 °C for 30 min, at applied pressures ranging from 125 to 500 MPa, and with different amounts of water. For temperatures lower than 160 °C, the obtained density was relatively low, about 70%, while the applied pressure had a minor effect. A high green density of about 83% was obtained by increasing the temperature up to 180 °C while applying 375 MPa of pressure in the presence of the 5 wt. % of water. After cold sintering, a post-annealing treatment was performed at 1200 °C for 10 h. In this way, BCZY electrolytes with densities of up to 94% were produced, showing a remarkable proton conductivity of 2.5 × 10^−2^ S cm at 600 °C. The authors associated the high total conductivity with increased grain boundary conduction compared to traditionally sintered samples. The addition of higher amounts of water during cold sintering (10–20 wt. %) was found to detrimentally affect the densification and conductivity of the material due to the formation of a larger number of impurities, such as BaCO_3_ at the grain boundaries, that remain after post-annealing.

Kindelmann et al. [113] studied the possibility of applying cold sintering to densify BaCe_0.2_Zr_0.7_Y_0.1_O_3−δ_ ceramics by controlling the phase composition of the starting powders. Here, ceramic precursors optimized for solid-state reactive sintering composed of BaCO_3_, CeO_2_, ZrO_2_, and Y_2_O_3_ pre-calcined at 1100 °C for 1 h were added with 0.5 wt% of NiO, recalcined at 1300 °C for 1 to 20 h, planetary milled in ethanol, and finally used as the starting mixture for the cold sintering experiments after sieving at 100 µm. Ceramic powders were moisturized with 5 wt. % of water and treated at 350 °C and 400 MPa for 5 min in a FAST/SPS apparatus. The authors found that a higher green density of about 80% was obtained with a shorter calcination treatment of 1–5 h since, in these conditions, the starting powders are composed of a Ce-rich and a Zr-rich BCZY phase. Longer calcination times led to the formation of the target pure-phase BCZY, decreasing the cold sintering activity. For BCZY-type materials, the driving force toward dissolution and precipitation events during cold sintering was found to be favored by the use of a Ce-rich BCZY phase as the starting powder. Water preferentially dissolves barium from the perovskite structure, followed by cerium and yttrium, leading to the formation of a Zr-rich BCZY phase, barium hydroxide, and ceria/Y-doped cerium oxide compounds, as follows:(13)$$\left(Ce\;rich\right)Ba\left(Ce,Zr,Y\right)O_{3}+H_{2}O\;\left(Zr\;rich\right)Ba(Ce,Zr,Y)O_{3}+Ba(OH{)}_{2}+Ce(Y)O_{2}$$

As explained above, BaCO_3_ is also produced as a consequence of reaction (9). These byproducts that formed during cold sintering had low strength, easing particle rotation and sliding under high mechanical pressures, enabling densification [149]. However, a post-annealing step was always required to induce solid-state reactions between impurities to obtain highly dense and phase-pure ceramics. After a post-annealing step at 1300 °C for 10 h, a phase-pure BCZY electrolyte showing a 96% relative density was obtained with nanometric grain size and an exceptional proton conductivity of 4 × 10^−2^ S cm at 600 °C, which is among the highest reported in the literature for BCZY-type proton conductors [25]. This is related to the unique characteristics offered by the cold sintering treatment: as the process undergoes dissolution and precipitation reactions, secondary phases such as BaCO_3_ and Y-doped CeO_2_ are produced in the grain boundary (GB) region, and during the post-annealing treatment, they react together with the BCZY phase, changing the grain boundary composition (Figure 5a,b). This leads to cationic segregations (and enrichment) in the GB region (Figure 5c–e), which causes a huge drop in the grain boundary resistivity and, consequently, an increase in total conductivity.

**Table 3 materials-17-05116-t003:** Cold sintering parameters and related post-annealing (PA) conditions for the processing of barium cerate and/or zirconate ceramics reported in the literature: composition, particle size (d_50_), the type and amount of transient phase, temperature (T), heating rate (R), pressure (P), dwell time (t), the relative density of the as-cold-sintered and post-annealed samples, and the total conductivity values (σ_T_) of the post-annealed specimens, measured in wet air at 600 °C.

Material	Starting Powders’ Characteristics		Transient Phase		Cold Sintering Parameters				PA		Density (%)		σ_T,600°C_ Wet Air (S/cm)	Ref.
	Composition	d_50_ (µm)	Type	wt%	T (°C)	R (°C/min)	P (MPa)	t (h)	T (°C)	t (h)	CSP	PA		
BaZr_0.8_Y_0.2_O_3−δ_	BaZr_0.8_Y_0.2_O_3−δ_ synthetized by a modified Pechini method [200]	n.r.	3 wt % PVA_(aq)_	20	180	10	400	1	1500	12	76	94	0.003	[112]
^$^ BaZr_0.8_Y_0.2_O_3−δ_	Mixture of BaCO_3_, ZrO_2_, and Y_2_O_3_ ball milled in isopropanol and calcined at 1175 °C for 3 h	<0.5	H_2_O	5	250	20	400	0.1	1100 ^#^	0.1	87	95	1 × 10^−5^	[148]
* BaZrO_3_	Ba(OH)_2_·8H_2_O	0.5	Structural water		350	20	500	2	/		92	/	n.r.	[199]
	Zr(OH)_4_	0.02–0.1												
	Ba/Zr = 1.15	/												
BaCe_0.8_Zr_0.1_Y_0.1_O_3−δ_	BaCe_0.8_Zr_0.1_Y_0.1_O_3−δ_ obtained by nitrate–glycine method [201]	23.3	H_2_O	5	180	5	375	0.5	1200	10	83	94	0.025	[110,111]
BaCe_0.7_Zr_0.2_Y_0.1_O_3−δ_	Commercial BaCe_0.7_Zr_0.2_Y_0.1_O_3−δ_ with bimodal particle size (Cerpotech)	0.8–2; 5–11	H_2_O	5	180	5	375	1	1600	15	n.r.	91	0.004	[150]
^$^ BaCe_0.2_Zr_0.7_Y_0.1_O_3−δ_	Mixture of BCY, BZY, BaCO_3_, and (Zr,Y)O_2_ + 0.5 wt. % NiO [202] calcined at 1300 °C for 5 h, milled, and sieved at 100 µm	0.6	H_2_O	5	350	20	400	0.1	1300	10	80	96	0.04	[113,149]

n.r. = not reported; ^$^ = cold sintering performed in a FAST/SPS facility; * = reactive cold sintering; ^#^ = PA performed in the same FAST/SPS setup applying 100 MPa. of pressure.

As previously mentioned, the composition and characteristics of the starting powders (size, morphology, etc.) can drastically influence the densification degree during cold sintering (and PA) and the final electrochemical properties. For example, Castellani et al. [150] obtained ≈91% dense BaCe_0.7_Zr_0.2_Y_0.1_O_3−δ_ ceramics after cold sintering commercial BCZY powders at 180 °C and 375 MPa for 60 min in the presence of 5 wt. % of water, followed by a post-annealing treatment at 1600 °C for 15 h. Similar or even better results in terms of densification degree and electrochemical properties can be obtained with conventional approaches (Table 1), as the proton conductivity was found to be one order of magnitude lower compared to traditionally sintered devices. These studies clearly pointed out that the composition of the starting ceramic powders has to be carefully evaluated to properly trigger cold sintering reactions and tune the grain boundary composition of the material.

Achieving pure and dense BCZY-based ceramics under mild conditions is the primary requirement for cold sintering to be considered a viable method for processing proton-conductive ceramics. As explained in the Introduction, this class of materials is widely used in proton-conductive solid oxide cells and electrolyzers, which are multilayer-structured devices. The only study regarding the feasibility of performing cold sintering in the production of anode-supported proton-conductive half-cells was recently reported by Kindelmann et al. [203]. In this case, the die was filled in a multi-step manner, pouring first a thicker layer (≈2.0 mm) of the composite BaCe_0.2_Zr_0.7_Y_0.1_O_3−δ_-NiO (BCZY:NiO = 1:1) powders and then a thinner film of pure BCZY (≈0.7 mm). The bilayer was then pre-compacted, and the disc was humidified with 5 wt% of water before being cold-sintered at 350 °C and 400 MPa for 5 min in a FAST/SPS setup. In these conditions, the dissolution of NiO was negligible, and cold sintering was promoted by the BCZY phase, as explained above, leading to a green density of up to 80%. The half-cells were then post-annealed at 1300 °C for 10 h in air and subsequently treated at 900 °C in H_2_/Ar to reduce NiO to metallic nickel (Figure 6).

A good contact between the two layers and a high densification degree were obtained; however, the microstructure of the composite needs further optimization. Moreover, the thickness of the dense electrolyte layer was ≈0.3 mm thick, and further work should be carried out to produce thin layers of ≈0.02 mm that can be effectively applied in protonic solid oxide cells.

## 6. Conclusions and Future Perspective

Cold sintering is undeniably one of the most promising techniques for obtaining dense ceramic materials with tailored functionalities in mild conditions. Thanks to the ease of operation, material flexibility, low cost, and high microstructural control, cold sintering is a very promising route to process ceramic materials such as doped barium cerate and zirconate proton conductors, which present high-temperature instabilities coupled with poor sinterability. The studies conducted up to now, especially by Kindelmann and co-workers [113,148,149,203], have fully demonstrated the possibility of applying cold sintering for the production of proton-conducting ceramics, laying the foundation for future developments. Cold sintering is actually performed as a pre-treatment to engineer the grain boundary composition of BCZY-based ceramics through the precipitation of secondary phases such as Ba(OH)_2_, CeO_2_, and Y_2_O_3_, which are formed as a consequence of the incongruent dissolution of BCZY-type materials in a water environment. Therefore, a subsequent post-annealing treatment is generally required to recover the perovskite phase. Currently, the substantial advantage that can be obtained from the cold sintering of BCZY-based ceramics using water as a sintering aid is the possibility to produce highly dense and conductive materials at a reduced temperature of about 1200–1300 °C. However, before applying cold sintering for the production of PC-SOC/SOE, several aspects have to be clarified and optimized, including those listed below:

(i)A deep understanding of the cold sintering mechanism is required to assess the exact types and amounts of species produced during the process and optimize the reproducibility of the experiments. In this context, ReaxFF molecular dynamics simulations were found to be a powerful tool for understanding and predicting the mechanisms involved during cold sintering [204,205,206]. Moreover, the in situ monitoring of material evolution during sintering using small-angle X-ray scattering [192,207,208], acoustic wave speed and attenuation measurements [209], electrochemical impedance spectroscopy [210,211,212], or a combination thereof was proven to be effective in understanding microstructural changes and defect formation during cold sintering, giving further insight into the cold sintering mechanism. It is worth mentioning the recent work published by Maier [158], who built a custom cold sintering device to densify materials in capillary tubes and monitored the densification through piston displacements and cameras, facilitating the study and optimization of cold sintering kinetics.(ii)The fabrication of larger samples with different geometries is necessary since most of the cold sintering experiments have used small discs with 8–13 mm diameters. The process scale-up is always tricky, but considering that cold sintering instrumentation is similar to that used in dry pressing and FAST/SPS technologies, it should not be too challenging from an engineering perspective. However, the enlarged sample size will open a new domain of processing challenges related to the homogeneity of the transient phase–ceramic mixture [213].(iii)Dense and flat thin films with a thickness of less than 0.1 mm have never been proven for this class of materials. In this regard, coupling standard thin-film manufacturing technology, such as tape casting and screen printing, with cold sintering would probably solve this issue, as already demonstrated for PbZrTi (PZT) ceramics [214], batteries [215,216], and microelectronic devices [79,217,218].

Moreover, since only water has been used as the transient phase to cold-sinter BCZY-type ceramics, other strategies to induce pressure solution creep mechanisms should be explored to properly tackle the cold sintering process and obtain high-density and phase-pure ceramics in one step. These include the use of a cation-rich water-based solution (Ba^2+^-Ce^4+^-Zr^4+^) to control the dissolution behavior of the perovskite structure, or the use of non-aqueous solutions such as solvents, hydrated salts, and eutectic mixtures. It is worth mentioning that the proper optimization of the transient phase employed during cold sintering was found to be the key step in cold sintering several types of materials, such as BaTiO_3_, PZT, and all-solid-state lithium batteries, leading to high-density and efficient materials and devices in one step.

## Figures and Tables

**Figure 1 materials-17-05116-f001:**
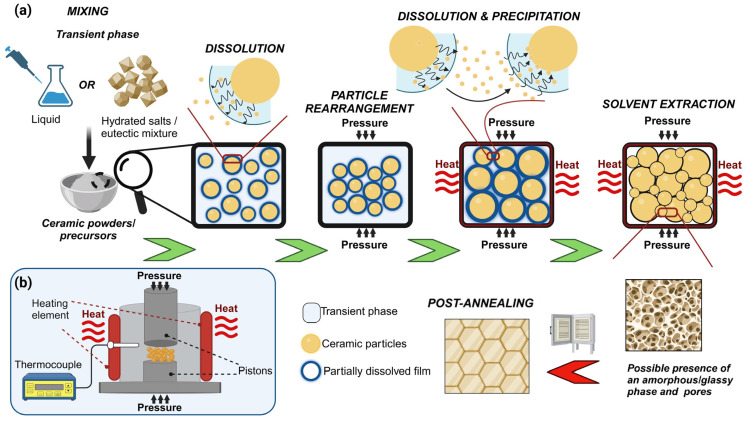
Schematic representation of the cold sintering process (**a**) and the thermocompression apparatus (**b**).

**Figure 2 materials-17-05116-f002:**
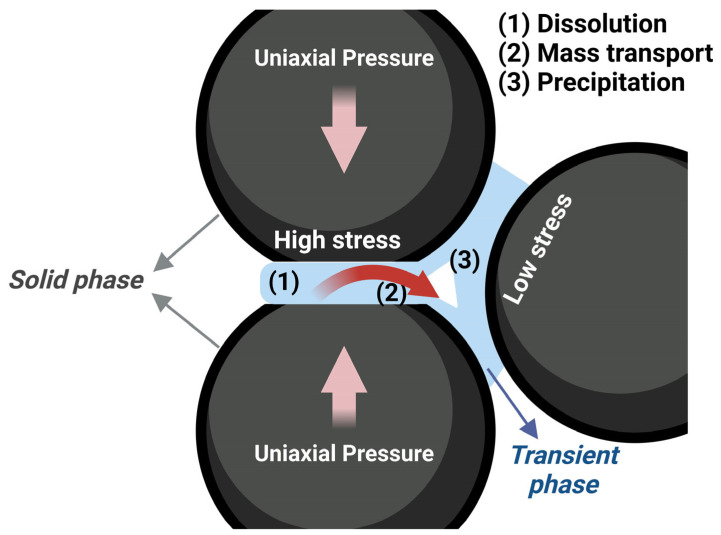
Schematic representation of mechanical–chemical effects during cold sintering process.

**Figure 3 materials-17-05116-f003:**
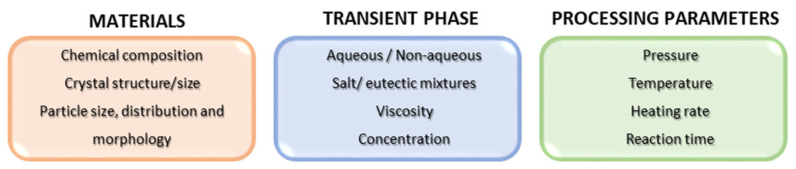
Parameters affecting the cold sintering process [127].

**Figure 4 materials-17-05116-f004:**
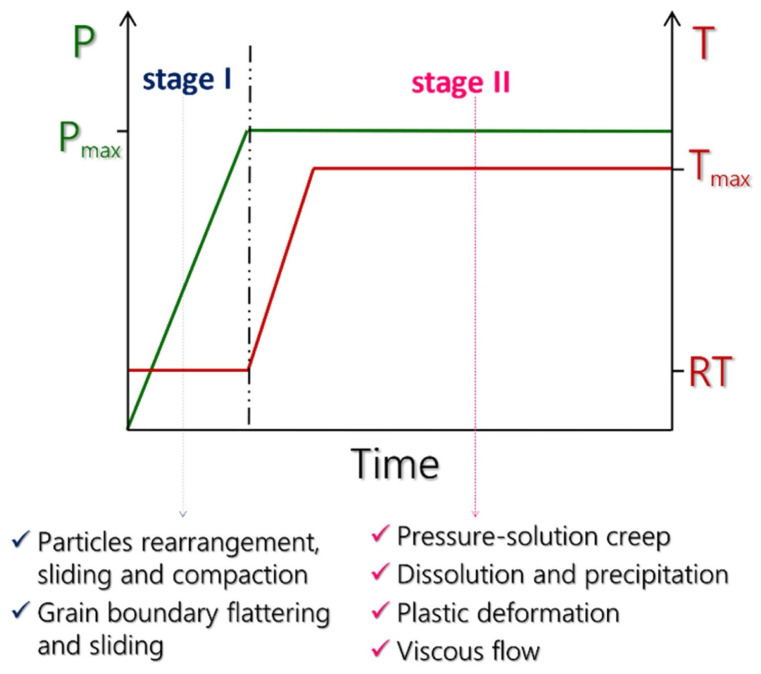
A schematic representation of the stages involved in the cold sintering process [152].

**Figure 5 materials-17-05116-f005:**
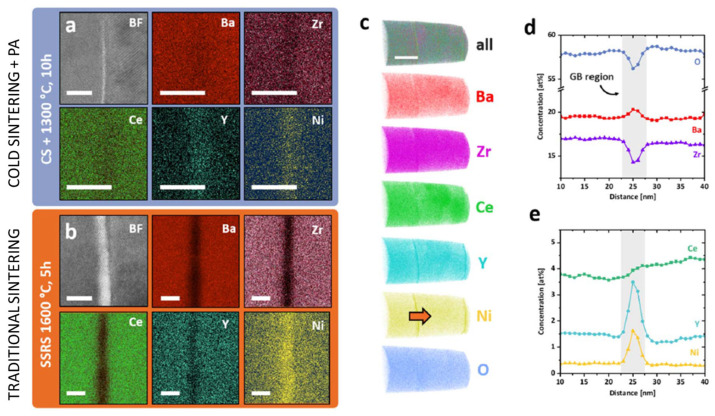
STEM micrographs and corresponding EDS maps (the scale bars are 5 nm) of a GB in a sample produced by cold sintering + PA (**a**) and conventionally sintered (**b**). Atomic Probe Tomography investigation of grain boundaries (scale bar is 50 nm) in the cold-sintered +PA sample (**c**) and the relative composition profile (orange arrow in (**c**)) of a random GB (**d**,**e**) [149].

**Figure 6 materials-17-05116-f006:**
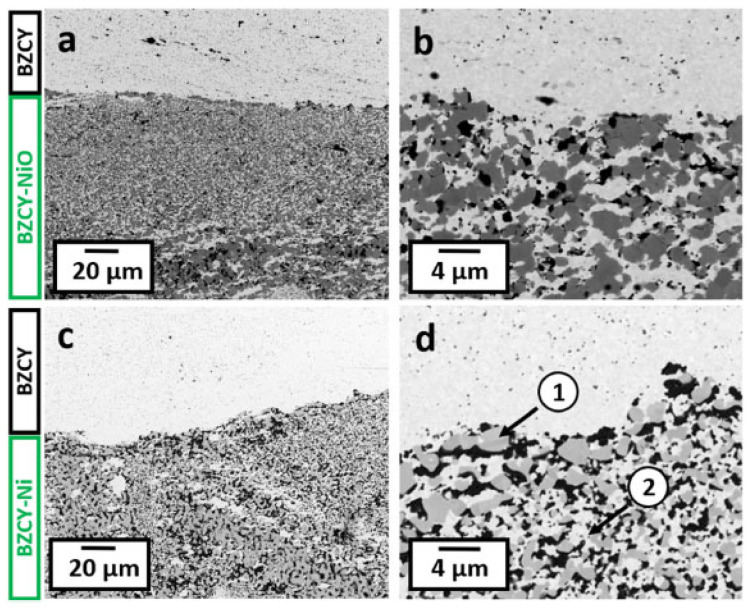
Microstructures of the BZCY-BZCY/NiO half-cell after cold sintering and post-annealing (**a**,**b**) and after subsequent reduction treatment (**c**,**d**); detail 1 shows the reduced Ni metal structure and detail 2 the remaining intact BZCY network [203].

## Data Availability

No new data were created or analyzed in this study.

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
