# Peer review of "Exploring the Potential of Cold Sintering for Proton-Conducting Ceramics: A Review"

_materials, 2024, doi:10.3390/ma17205116_

Round 1
Reviewer 1 Report
Comments and Suggestions for Authors
This manuscript focused on the cold sintering for Proton-conducting ceramics, such as BCZY-type materials. The manuscript provide a detailed exposition and comparative analysis of the research background, the opportunities and challenges. To obtain fully dense ceramic in a lower temperature, the cold sintering technology should be a potential technological solution. I suggested the publication of this manuscript after minor revision, the questions are as followed:
1. In the abstract part, line 10 and 13, the authors should check and unify the correct abbreviations of BZCY.
2. The manuscript describes inconsistent dissolution as an impediment to densification, which can be mitigated or avoided by the use of cations with higher solubility in selected solvents or by the addition of molten hydroxides, salts, chelates, etc., and if additives cannot be used in the case of ceramic powders with high purity requirements, are there any other ways of avoiding this phenomenon? Any explanation or suggestions?3. It is stated in the article that pressure is one of the important factors affecting the cold sintering process, and that there exists a critical value of pressure, below which the applied pressure enhances densification and grain growth, while above which densification is significantly delayed, how can the size of the critical value be determined for different materials?
4. When considering the crystal structure of a material, how does the crystal structure affect the choice of transient phase? For ceramic materials doped with different elements, how does the type and content of the dopant element affect the choice of transient phase?
Author Response
Dear reviewer, please find below the point-by-point response to your comments. Each answer to a comment Is addressed "Response" and written in red. Where we believe it is applicable, we have adjusted the manuscript, the details of which are given in the responses and within a version of the text with changes highlighted in yellow.
Comment 1: In the abstract part, line 10 and 13, the authors should check and unify the correct abbreviations of BZCY.
Response 1: We thank the reviewer for the suggestion, we have unified the abbreviations of BaCexZryY1-x-yO3-δ as BCZY.
Comment 2: The manuscript describes inconsistent dissolution as an impediment to densification, which can be mitigated or avoided by the use of cations with higher solubility in selected solvents or by the addition of molten hydroxides, salts, chelates, etc., and if additives cannot be used in the case of ceramic powders with high purity requirements, are there any other ways of avoiding this phenomenon? Any explanation or suggestions?
Response 2: We thank the reviewer for the comment. When incongruent dissolution occurs there are three main possibilities to obtain a phase pure material:
- The most common way to overcome this issue is to perform a post annealing treatment after cold sintering to induce solid state reactions between the secondary phases formed during cold sintering.
- Another way, is to tailor the composition of the transient phase depending on the dissolution behaviour of the elements present in the ceramic material to densify. For example, aqueous solutions are able to densify perovskites like BaTiO3 or SrTiO3, even if they incur in incongruent dissolution: typically, Ba2+ or Sr2+ preferentially leaches out and dissolves in water while an amorphous TiO2 layer forms and wraps around the inner part of the particle, limiting the densification. In these cases, the addition of Ba(OH)2/TiO2 or SrCl2/TiO2 nanoparticles to water for BaTiO3 and SrTiO3 respectively, was found to limit the formation of the insoluble TiO2 layer around the particles, increasing the densification rate during cold sintering, achieving highly pure ceramics.
- Changing the nature of the transient phase, using for example alkali metal hydroxide fluxes such as NaOH-KOH 51:49 eutectic mixture (NaK). High-dense (>96%), pure BaTiO3 ceramics were obtained by Tsujia et al. (Journal of the European Ceramic Society 40 (2020) 1280–12) using 6 wt. % of NaK eutectic flux under 520 MPa and 300°C for 12 h. During the densification, complex dynamics occur between Na+, K+, and the BaTiO3 surface, with the formation of transient complexes that enable the dissolution of both Ba and Ti from the bulk material, and the subsequent densification through the pressure solution creep mechanism. In some cases, secondary phases were formed due to Na-Metal interactions but generally, they can be eliminated by a simple washing step.
In general, the key point is to find the right transient phase that promotes congruent dissolution and limits the formation of secondary phases. In this regard, predominance and existence diagrams can help with the selection of the transient phase.
Comment 3: It is stated in the article that pressure is one of the important factors affecting the cold sintering process, and that there exists a critical value of pressure, below which the applied pressure enhances densification and grain growth, while above which densification is significantly delayed, how can the size of the critical value be determined for different materials?
Response 3: We thank the reviewer for the comment, the role of the pressure during cold sintering is quite different from traditional sintering. The critical pressure value discussed in the manuscript was experimentally determined for ZnO by Y. Shi et al. (Ceramics International 48 (2022) 30517–30523) who performed several cold sintering experiments at different applied pressures from 0.05 to 2 GPa. They found that densification and grain growth increased linearly with increasing the applied pressure up to 1 GPa, while lower density and grain size values were obtained for higher pressures. The authors associated this behaviour with the formation of a very high-stress gradient in the environment when the pressure is above the critical value, and homogeneous nucleation is induced in the liquid phase instead of recrystallization and grain growth.
To conclude, the critical pressure value was demonstrated and determined only for ZnO which is the model material used to study the cold sintering process. We believe that the systematic study of the cold sintering processing parameters can be an effective way to understand the size of the critical pressure value for different materials. This can be done experimentally, performing cold sintering experiments at increasing applied pressure as reported by Y. Shi et al.
Comment 4: When considering the crystal structure of a material, how does the crystal structure affect the choice of transient phase? For ceramic materials doped with different elements, how does the type and content of the dopant element affect the choice of transient phase?
Response 4: We thank the reviewer for pointing out these questions. We will answer separately at the two questions:
- When considering the crystal structure of a material, how does the crystal structure affect the choice of transient phase?
The crystal structure of the starting ceramic powders can influence the densification behavior during cold sintering since dissolution and precipitation reaction rates can be influenced by the crystal planes exposed (Nano Lett. 2021, 21, 3451−3457; ACS Appl. Mater. Interfaces 2018, 10, 37717−37724). Moreover, allotropic transitions can also be triggered during cold sintering (Rev. Sci. Instrum. 90, 055104 (2019); Journal of the European Ceramic Society 44 (2024) 2754–2761). Therefore, there is not a direct correlation between crystal structure and the choice of the transient phase, but is something to be taken into account, especially if allotropic transitions can occur.
- For ceramic materials doped with different elements, how does the type and content of the dopant element affect the choice of transient phase?
It depends on how the type and level of doping influence the solubility of the material in the selected transient phase. The ideal case is when a congruent dissolution occurs between the ceramic material and the transient phase, scilicet the material dissolves into the transient phase with a homogeneous chemical stoichiometry before mass transport and precipitation. When a dopant is introduced in the system, it may change the dissolution behaviour of the material leading to possible incongruent dissolution and the formation of unwanted secondary phases. Therefore, it’s important to study the behaviour of each element/dopant in the selected transient phase.
Reviewer 2 Report
Comments and Suggestions for Authors
In this manuscript, the authors provided a comprehensive review on the use of a promising approach called “cold sintering” for fabricating proton-conducting ceramics. It offered insights into the opportunities as well as challenges of this method for developing proton-conducting ceramics for use in protonic ceramic electrochemical cells. This work is a timely contribution to this field. The manuscript was well organized and presented. I therefore believe it is a good fit for the journal Materials. However, to further enhance the manuscript’s clarity, I feel some certain revision to address the comments listed below is needed.
1. The discussion of proton conducting ceramics was often made under the context of protonic ceramic fuel cells or electrolysis cells (PCFC/PCEC). Are there examples on the cold sintering prepared ceramics that have applications in PCFC/PCEC? Including such examples in the text might further add to the promise of this method for use in PCFC/PCEC.
2. To appeal to a broader readership, recent works on PCFC/PCEC are recommended to be referenced in the Introduction section (e.g., Mater. Horiz., 2020, 7, 2519; Small, 2021, 17, 2101872).
3. Line 208, the authors had a sub-section of “3.1” (3.1. Cold Sintering mechanisms), however, there is no main section of “3”. Therefore, this should probably be section 3 and needs revision for the section numbering.
4. Figure 3, for the very left side Materials section, please note that “Chemical omposition” is a typo and needs revision.
5. Some of the technical terms, such as “transient phase”, should be properly defined at their first appearance such that general readers would be able to understand.
6. Figure 5c, for the APT images, would there be a scale bar for these images? I noticed there is a white scale bar on the very top of the images, but its scale was not identified.
Author Response
Dear reviewer, please find below the point-by-point response to your comments. Each answer to a comment Is addressed "Response" and written in red. Where we believe it is applicable, we have adjusted the manuscript, the details of which are given in the responses and within a version of the text with changes highlighted in yellow.
Comment 1: The discussion of proton conducting ceramics was often made under the context of protonic ceramic fuel cells or electrolysis cells (PCFC/PCEC). Are there examples on the cold sintering prepared ceramics that have applications in PCFC/PCEC? Including such examples in the text might further add to the promise of this method for use in PCFC/PCEC.
Response 1: We thank the reviewer for the comment. Cold sintering is a relatively new technology and just few works have been published in the field of proton conductive ceramics or more in general, in the PCFC/PCEC field. To date, the fabrication of a complete PCFC/PCEC assembly through the application of the cold sintering technique have never been proved. The studies conducted in this field were mainly devoted on the production of BZY/BCZY – based proton conductive electrolytes. Recently, the cold sintering of a BCZY-NiO/BCZY anode-supported half-cell have been published by M. Kindelmann et al. (Ceramics International 50 (2024) 37373–37378). To the best of our knowledge, all the case studies reported in literature are present in the manuscript and described in section 5 “Application of the cold sintering process to ceramic proton conductors”.
Comment 2: To appeal to a broader readership, recent works on PCFC/PCEC are recommended to be referenced in the Introduction section (e.g., Mater. Horiz., 2020, 7, 2519; Small, 2021, 17, 2101872).
Response 2: We thank the reviewer for the suggested references, we have added the latter (Small, 2021, 17, 2101872) to the manuscript in the Introduction section (page 1). The first suggested reference (Mater. Horiz., 2020, 7, 2519) is an interesting review on the use of Ruddlesden–Popper perovskites in electrocatalysis. However, we believe it falls somewhat outside the scope of our manuscript, as our focus is specifically on BCZY-type proton-conducting perovskites.
Comment 3: Line 208, the authors had a sub-section of “3.1” (3.1. Cold Sintering mechanisms), however, there is no main section of “3”. Therefore, this should probably be section 3 and needs revision for the section numbering
Response 3: We thank the reviewer for the comment, we have corrected the numbering of section 3.
Comment 4: Figure 3, for the very left side Materials section, please note that “Chemical omposition” is a typo and needs revision
Response 4: We thank the reviewer for their remark and apologize for the oversight. We have corrected Figure 3 and uploaded the revised version accordingly.
Comment 5: Some of the technical terms, such as “transient phase”, should be properly defined at their first appearance such that general readers would be able to understand.
Response 5: We thank the reviewer for this suggestion, we have modified the text of Section 2 Cold sintering process as follows (page 5):
“During cold sintering, materials are densified in the presence of a transient liquid phase, which facilitates dissolution-precipitation reactions and promotes compaction under externally applied uniaxial pressure (100–1000 MPa). Consolidation occurs at temperatures ranging from room temperature to just above the liquid's boiling point (typically < 350 °C).”
Comment 6: Figure 5c, for the APT images, would there be a scale bar for these images? I noticed there is a white scale bar on the very top of the images, but its scale was not identified.
Response 6: We thank the reviewer for the comment. In Figure 5 the scale bar of the STEM/EDX micrographs (Figure 5a,b) and atomic probe tomography APT (Figure 5c) are described in the figure caption as follow:
“STEM micrographs and corresponding EDS maps (the scale bars are 5 nm) of a GB in a sample produced by cold sintering + PA (a) and conventionally sintered (b). Atomic Probe Tomography investigation of grain boundaries (scale bar is 50 nm) in the cold sintered +PA sample (c) and relative composition profile (orange arrow in c) of a random GB (d,e)”
Reviewer 3 Report
Comments and Suggestions for Authors
The review article "Exploring the Potential of Cold Sintering for Proton-Conducting Ceramics: A Review" is of interest in the field of ceramic processing of various materials with applications in alternative energy. The discussion of the parameters that influence processing, as well as their advantages compared to traditional methods, highlights the importance of its publication in guiding researchers in future work. It would be important to consider the following points, if possible:
-
It is important to provide a brief description of the perovskite structure.
-
It would be interesting, if possible, to present the compositional limits for the formation of solid solutions for BaCexZr1-xYyO3-d. A phase diagram could be helpful.
Author Response
Dear reviewer, please find below the point-by-point response to your comments. Each answer to a comment Is addressed "Response" and written in red. Where we believe it is applicable, we have adjusted the manuscript, the details of which are given in the responses and within a version of the text with changes highlighted in yellow.
Comment 1: It is important to provide a brief description of the perovskite structure.
Response 1: We thank the reviewer for this suggestion and agree it’s important to describe the perovskite structure. We have added the following text in the Introduction section (page 1-2)
Perovskites oxides are among the most promising ceramic proton conductors. Ideal perovskites have the general formula ABO3, where the A-sites are typically occupied by larger cations than the B-site and analogous in size to the O-site anions. ABO3 structure can be considered as a face centered cubic (fcc) lattice with the A atoms at the corners and the O atoms on the faces. The B atom is located at the center of the lattice. The structure of perovskite is formed by a three dimensionally connected system of BO6 octahedra and a AO12 cube-octahedra at the edges of the B-centered cubic lattice. In proton conductive perovskites, a trivalent cation is generally partially substituted in the B site to increase the oxygen vacancy concentration in the composite. The general formula of these doped perovskites can be written as AB1−xMxO3−δ, where x is less than the upper limit of its solid solution formation range (usually less than 0.2) and δ denotes the number of oxygen deficiencies per unit formula of the perovskite.
Comment 2: It would be interesting, if possible, to present the compositional limits for the formation of solid solutions for BaCexZr1-xYyO3-d. A phase diagram could be helpful.
Response 2: We thank the reviewer for this suggestion. To the best of our knowledge, no phase diagrams are present in the literature regarding Y-doped BaCeO3 – BaZrO3 solid solutions. K. Ueno et al. (Journal of Solid State Electrochemistry (2020) 24:1523–1538) reported the ternary BaO-ZrO2-Y2O3 phase diagram to study the compositional limits of BaZr1-xYxO3-δ ceramics, however a quaternary BaO-CeO2-ZrO2-Y2O3 phase diagram is not available. It’s difficult to state what is the exact compositional limit in BaCe1−x-yZrxYyO3-δ ceramics, however the compositions generally considered for PCFC/PCEC applications are those with x ≤ 1 and 0.1 ≤ y ≤ 0.2.
Since the focus of the manuscript is on the sintering behaviour of BCZY-type ceramics, we believe that calculating and building an ad hoc phase diagram for the BCZY system will fall a bit outside the scope of the review.
P.S. Revisiting this point I noticed that in the text (page 2, line 51) and Table 1, the term BaCexZr1-xYyO3-d was not unified, therefore we have modified the description of Table 1 and the text using the term BaCexZryY1-x-yO3–d.